# Biopsychosocial Strategies for Alleviating Low Back Pain in Late Mothers: A Systematic Review

**DOI:** 10.3390/healthcare13111237

**Published:** 2025-05-23

**Authors:** Darshika Thejani Bulathwatta, Małgorzata Treppner, Monika Januszkiewicz, Paulina Głowacka, Judyta Borchet, Asanka Bulathwatta, Mariola Bidzan

**Affiliations:** 1Department of Psychology and Counseling, Faculty of Health Sciences, The Open University of Sri Lanka, Colombo 11222, Sri Lanka; 2Institute of Psychology, Faculty of Social Sciences, University of Gdansk, 80-309 Gdansk, Poland; m.treppner.196@studms.ug.edu.pl (M.T.); m.januszkiewicz.171@studms.ug.edu.pl (M.J.); judyta.borchet@ug.edu.pl (J.B.); mariola.bidzan@ug.edu.pl (M.B.); 3Academic Center for Psychological Support, University of Gdansk, 80-309 Gdansk, Poland; 4Institute of Pedagogy and Languages, University of Applied Sciences in Elbląg, 82-300 Elbląg, Poland; 21162@student.ans-elblag.pl; 5Department of Psychology, Faculty of Arts, University of Peradeniya, Peradeniya 20400, Sri Lanka; asankab@arts.pdn.ac.lk

**Keywords:** low back pain, older mothers, biopsychosocial factors, biopsychosocial model

## Abstract

**Background:** Low back pain (LBP) in older mothers is a prevalent and multifaceted condition influenced by a range of biopsychosocial factors. As the trend of late motherhood increases globally, it is essential to understand how LBP affects this population from a biopsychosocial perspective. **Objective:** This systematic review aims to examine the biopsychosocial determinants of LBP in older mothers by synthesizing findings from quantitative studies published between January 2010 and May 2024. Specifically, it explores the biological, psychological, and social factors contributing to LBP in this population and how it affects their daily lives and well-being. **Methods:** A comprehensive literature search was conducted across PubMed, PsychInfo, Web of Science, EMBASE, DARE, and the Cochrane Library. Studies were screened for eligibility based on predefined criteria. Five quantitative studies with a combined sample of 118,964 participants were included. The methodological quality was assessed, and data were extracted for analysis. **Results:** All five studies addressed biological aspects of LBP, including pelvic girdle pain, hemorrhoids, and varicose veins. Three studies also explored psychological factors such as depression and stress. Two studies incorporated social dimensions, including inadequate support systems, occupational burdens, and healthcare access barriers. Age and parity were consistently reported as exacerbating physical and psychological symptoms. **Conclusions:** This review highlights the importance of considering biopsychosocial factors when managing LBP in older mothers. Tailored interventions, such as exercise programs, family support, and workplace accommodations, are essential for improving outcomes. Further research using longitudinal studies is needed to explore these factors in greater depth.

## 1. Introduction

Low back pain (LBP) and its related disabilities impose a substantial personal burden. Additionally, LBP causes significant personal suffering globally, and those with ongoing, debilitating symptoms contribute to considerable societal costs through healthcare expenses and decreased work productivity [1,2]. LBP is the primary cause of disability globally, with one in six Australians reporting back issues in 2017–2018. A study by Wu and colleagues (2020) revealed that in 2017, Southern Latin America reported the highest prevalence of low back pain (LBP) at 13.47%, followed closely by high-income Asia Pacific at 13.16% [3]. In contrast, East Asia had the lowest prevalence at 3.92%, followed by Central Latin America at 5.62%. In terms of the number of individuals affected, South Asia had the highest burden with 96.3 million cases, followed by East Asia with 67.7 million. The lowest numbers were observed in Oceania (0.7 million) and the Caribbean (2.7 million). According to the World Health Organization [4], low back pain (LBP) refers to discomfort located between the lower ribs and the buttocks andit may be short-term (acute), moderately prolonged (sub-acute), or persist over a longer duration (chronic). The global incidence of low back pain has been on the rise since 1990, driven by population growth and aging, particularly affecting individuals aged 40 to 80 [5]. Importantly, one demographic increasingly represented within this aging population is women who become mothers at a later age. Late motherhood—typically defined as pregnancy after the age of 35—has become more prevalent in many high-income countries, aligning with the same timeframe in which LBP has increased globally. As maternal age rises, so does the physiological strain during and after pregnancy, including musculoskeletal changes that can lead to low back pain. A significant source of low LBP can be the facet joints (or zygapophyseal joints), primarily due to inflammation, degenerative or arthritic changes, associated muscle disorders, or repetitive injuries [6]. Non-pharmacological care is regarded as a primary treatment option. Individuals with LBP frequently seek assistance from physiotherapists and other rehabilitation health professionals; however, optimal care typically involves an initial diagnosis and therapeutic plan developed by a rehabilitation physician or multidisciplinary team, followed by implementation through coordinated non-pharmacological interventions [2]. However, understanding and addressing LBP requires more than simply focusing on physical symptoms.

## 2. Late Motherhood

Late motherhood is defined as getting pregnant after the age of 35. It is a rising trend, especially in high-resource countries—for example, in the United States, where the number of births after the age of 35 rose by 64% between 1990 and 2008 [7]. The phenomenon is more common among women with higher education and of the middle class [8]. Delayed childbearing may be a conscious decision or not—it may occur because of waiting for the ‘perfect time’ [9], or because of problems with conception. In a study conducted by Martin in 2020 [10], the women interviewed listed career aspirations, finances, personal happiness, and stable relationships among some of the reasons for postponed motherhood. Another possible explanation for the growing number of late pregnancies might be the technological advancements in medicine, such as in vitro fertilization and donor eggs [11].

Late pregnancy is considered high-risk as the probability of gestational diabetes, chromosomal anomalies, pregnancy loss, preeclampsia, or hemorrhage is higher [12,13]. There are, however, social advantages of having a child at a later age—studies have shown that such children perform better at school [14]. Late mothers might struggle with social stigma—they are accused of putting their future children’s well-being and health ‘at risk’, or their parental and physical abilities are being questioned [11,13]. Another problem that older mothers may have to face is LBP. In a study from 2020, late pregnancy was recognized as a risk factor for pregnancy-related LBP [15]. Despite the established relevance of the BPS model in general LBP populations, little research has specifically focused on late mothers—a demographic facing unique biological, psychological, and social challenges. These interrelated factors create a complex pain experience that traditional biomedical models may fail to adequately address.

### 2.1. Biopsychosocial Model (BPS)

The adoption of the biopsychosocial model (BPS) in 1977 offered a framework to understand the intricate nature of disabling LBP and its multifaceted clinical reasoning, which persists to the present, and this model integrates the interplay between social, psychological, and biological aspects of pain alongside behavioral conditioning within the context of pain [5,16]. Melzack and Wall (1965) highlighted the development of the BPS model as a countermeasure to the limitations of the biomedical model, which has historically dominated Western healthcare, particularly in pain management [17]. Nevertheless, it is also increasingly recognized as important for understanding acute pain, where early psychosocial factors (such as anxiety, catastrophizing, or social isolation) can influence pain trajectories and outcomes. Current guidelines advocate for applying the BPS model in both acute and chronic stages of LBP, although its relevance and application may vary depending on the chronicity and complexity of the pain experience.

As Nisolle and Bourguignon (2023) pointed out, patients commonly perceive that the significance of tissue damage seen in imaging directly correlates with pain, yet psychological issues are frequently overlooked or disregarded [6]. Therefore, comprehensive management of chronic LBP that integrates this model is crucial, as it aids in breaking detrimental cycles (such as pain-induced sleep disturbances that lead to further pain). Guidelines suggest incorporating a biopsychosocial approach in evaluating and managing individuals with LBP, regardless of whether they are in the acute or chronic stages [2]. Willson and colleagues (2023) revealed that pain is experienced differently among individuals during pregnancy, and women scheduled for cesarean delivery—an increasingly common procedure—represent a relatively understudied group who may be at increased risk of pain. While biopsychosocial factors are known to influence various forms of chronic pain, their role in late pregnancy-related pain remains insufficiently explored [18]. Moreover, multidisciplinary teams in rehabilitation centers or specialized pain clinics often administer BPS interventions to patients experiencing moderate to high levels of disability. These interventions are more effective in reducing both pain and disability when compared to standard care [19].

### 2.2. Rationale

Low back pain (LBP) is a leading cause of disability worldwide, imposing significant personal and societal burdens through healthcare costs and decreased work productivity. Despite the prevalence of LBP, its complex nature often necessitates more than just traditional biomedical treatments. The biopsychosocial (BPS) model, which integrates biological, psychological, and social factors, provides a comprehensive framework for understanding and managing LBP.

Late mothers often face unique challenges, including higher medical risks and social stigma [20]. These factors can exacerbate the experience of LBP, making it essential to consider their specific biological, psychological, and social contexts in treatment approaches. The BPS model is particularly relevant for addressing LBP in late mothers [21]. It acknowledges the limitations of the biomedical model, which tends to overlook psychological and social dimensions of pain. By incorporating BPS strategies, healthcare providers can offer more effective, holistic care that addresses the root causes and perpetuating factors of LBP.

Given the rising prevalence of LBP and the increasing number of late mothers, it is imperative to explore and implement BPS strategies to alleviate LBP in this demographic. Therefore, this systematic review aims to consolidate current evidence on LBP in late mothers and its biopsychological determinants. By understanding the interplay of biological, psychological, and social factors, the review seeks to provide insights into comprehensive management strategies that can improve the quality of life for late mothers and reduce the broader societal impact of LBP. A systematic review conducted by Dionne (2006) points out that the prevalence of severe back pain has been found to increase with advancing age [22].

Thus, this systematic review explores how late motherhood is associated with LBP from a biopsychosocial perspective. The secondary research questions are:What biological, psychological, and social factors contribute to LBP in older mothers?How does LBP impact the daily lives and well-being of older mothers?

## 3. Method

The comprehensive literature search followed guidelines outlined in the Preferred Reporting Items for Systematic Review and Meta-Analysis Protocols (PRISMA-P) [23]. All the other parts of the systematic review and the final article were prepared based on the guidelines outlined in the Preferred Reporting Items for Systematic Review and Meta-Analyses (PRISMA). The protocol was registered in the International Prospective Register of Systematic Reviews (PROSPERO) and is available at https://www.crd.york.ac.uk/PROSPERO/ (accessed on 15 March 2025) under the registration number CRD42025646095.

### 3.1. Eligibility Criteria

Studies were selected according to pre-specified eligibility criteria following the PICOS model presented in Table 1. The search was not restricted to articles based on research conducted in a specific geographic location.

### 3.2. Search Strategy

#### 3.2.1. Information Sources

We conducted a comprehensive literature search using the following electronic databases: PubMed, PsychInfo, Web of Science, EMBASE, DARE, and the Cochrane Library. The search was performed up to 22 June 2024 and updated on 12 September 2024 using the same search strategy. We included studies published in English between 2010 and 2024 May and did not restrict by country or sample size. We excluded editorials, letters, case studies, case series, and conference abstracts.

#### 3.2.2. Search

The search strategy included terms relating to exposure (“older birth” OR “postponed motherhood” OR “geriatric pregnancy” OR “late-life pregnancy” OR “senior maternity” OR “delayed childbirth” OR “motherhood after 35”) and outcome (“low back pain” OR “lumbar pain” OR “lumbar spine pain” OR “nonspecific low back pain” OR “chronic low back pain”).

### 3.3. Study Selection

Four reviewers conducted database searches and manually examined the reference lists of potentially relevant articles. Any potentially relevant records were listed, and duplicates were identified and eliminated. Two review authors independently screened titles and abstracts to determine their eligibility based on the pre-defined criteria, documenting reasons for exclusion. Disagreements were resolved through discussions with another reviewer.

Full-text reports were obtained for studies that appeared to meet the inclusion criteria or where there was uncertainty. Two review authors independently assessed these full-text reports to determine if they met the inclusion criteria, providing justification for exclusions. A subset of reports was pre-tested against the eligibility criteria to ensure the robustness of the selection process.

The eligibility criteria were assessed in a prioritized order, beginning with participants, followed by the exposure of interest, comparator, outcome, and study design. If a study failed to meet a criterion, it was excluded at that point, and subsequent criteria were not evaluated. The primary reason for exclusion was recorded.

Study relevance was assessed by health psychology researchers with expertise in the content area. During the selection process, reviewers were not blinded to journal titles, study authors, or their institutional affiliations.

### 3.4. Data Extraction and Management

Data extraction was performed independently by two reviewers using a standardized data extraction form. Key information, including participant details, exposure of interest, comparator, outcomes, and study design, was systematically extracted. The extraction process followed a prioritized order, beginning with participant characteristics, followed by exposure, comparator, outcomes, and study design.

Any discrepancies in the data extraction process were resolved through discussions between the two reviewers, with a third reviewer consulted if necessary. All extracted data were carefully checked for accuracy and consistency.

A sample of the data extraction forms was pre-tested to ensure the clarity of the criteria and completeness of the extracted information. In cases where additional clarification was needed, the corresponding authors were contacted. The reasons for excluding any data were thoroughly documented.

### 3.5. Risk for Bias Assessment

In this systematic review, we used the JBI Critical Appraisal Checklist to assess the risk of bias in the included studies. This checklist provides a structured evaluation of the methodological rigor of each study. We considered various factors, including study design, sample selection, outcome measurement, statistical analysis, and potential sources of bias. Studies were categorized into risk levels (low, moderate, or high) based on the quality of their methodological features. For cohort studies, we assessed the clarity of the sampling process, the validity of outcome measures, the presence of confounding factors, and the use of statistical techniques to control for these confounders. Cross-sectional studies were evaluated based on the accuracy of outcome measurement, representativeness of the sample, and the clarity of the data collection process. The sample size reflects the scale and power of each study, indicating how well the findings can be generalized to the target population. The sampling method provides insight into how participants were selected, influencing the representativeness and potential biases of the sample. A larger sample size generally increases the statistical power of a study, while the sampling method determines the level of randomness and control over potential confounding variables.

## 4. Results

In this section, the authors briefly present the results of the SLR study.

### 4.1. Study Characteristics

The computer-aided database search yielded 65 records (Figure 1). After duplicate removal, 63 records remained and were screened based on title and abstract. As a result, eight records were selected for full-text assessment to determine eligibility. Ultimately, five articles met the predefined criteria and were included in the systematic review.

A total of five studies were included, varying in design, including prospective cohort, cross-sectional, and multicenter studies. These studies were conducted in different countries, including Iran, Norway, and the UK. The selection process and full Preferred Reporting Items for Systematic Reviews are presented in Figure 1 and Table 2.

### 4.2. Description of Studies

A summary of the selected articles included in this systematic review is presented in Table 3. All the studies were observational and utilized a quantitative methodology. Two studies employed a cross-sectional design (n = 2; [15,24]). The remaining studies employed a prospective cohort approach, including one prospective population-based cohort study (n = 1; [25]) and two prospective cohort studies (n = 2; [26,27]). The studies were published between 2010 and 2020.

### 4.3. Population Characteristics

The studies included in this review focused on pregnant and postpartum women, with a primary emphasis on older pregnant women (n = 118,964), as summarized in Table 2. One study [15] examined the prevalence of pregnancy-related low back pain (LBP) and its influencing factors across different stages of pregnancy among participants aged 18 years and older. Another study [27] investigated maternal health and exercise in women aged 35 and above, emphasizing its impact on advanced maternal age. The role of sports and physical activity in reducing pelvic girdle pain was explored in a cohort study [25]. Additionally, a study [26] assessed the association between parity and pelvic girdle syndrome (PGS) without age restrictions. Another study [24] compared self-reported pregnancy-related symptoms, health status, and the use of antenatal services among older women (aged 35 and above) with those in younger age groups (below 25 years and 25–34 years).

Biological factors contributing to low back pain in late mothers

All five studies consistently highlighted the role of biological factors in contributing to low back pain (LBP) among late mothers [15,23,24,25,26,27]. One study found that the prevalence of LBP increases as gestational age advances, with late mothers experiencing greater difficulty in tolerating pain compared to younger mothers [15]. Their study further emphasized that advanced maternal age is associated with a diminished ability to cope with pain, which becomes more pronounced in the later stages of pregnancy. Additionally, one study [15] reported that physiological changes, including increased lumbar curvature and alterations in lumbar positioning, contribute significantly to the onset of LBP. Furthermore, their findings indicated that obesity and high BMI, which are more common among older pregnant women, exacerbate lumbar strain, leading to heightened pain and discomfort.

A study found that the prevalence of pregnancy-related symptoms varied significantly across different age groups and by parity [24]. Older/late mothers were more likely to experience conditions such as varicose veins, hemorrhoids, carpal tunnel syndrome, and stress incontinence, which were often linked to previous pregnancies and childbirth. In particular, older multiparous women reported a higher incidence of varicose veins, hemorrhoids, and stress incontinence, highlighting the cumulative impact of multiple pregnancies on maternal health.

Interestingly, despite these physical discomforts, older mothers were significantly less likely than younger mothers to report symptoms such as repeated vomiting, back pain, and depression. This could be attributed to a greater acceptance of pregnancy-related changes, where older women may perceive these symptoms as a natural and inevitable part of the maternal experience. Additionally, this study [24] suggested that societal and medical perspectives may contribute to this underreporting, as both older mothers and healthcare professionals may consider these symptoms as ‘normal’, reducing the likelihood of seeking medical help.

Another study [25] found that younger women had a higher risk of pelvic girdle pain. Their results also suggest that older mothers were more likely to engage in regular exercise before pregnancy, which may have contributed to a lower prevalence of pelvic girdle pain in this group. Additionally, despite the protective benefits of exercise, high-impact workouts and frequent physical activity (up to five times weekly) remained associated with pelvic girdle pain, regardless of age and BMI. This finding highlights the complex interaction between maternal age, physical activity, and musculoskeletal health during pregnancy.

Similarly, one study [26] found that women who have had multiple pregnancies (multiparous women) experience increased joint mobility in their pelvis compared to women who have never been pregnant (nulliparous women). This is due to hormonal changes during pregnancy. This extra mobility, combined with mechanical strain, can contribute to pelvic girdle pain (PGP). The study also showed that women who have had multiple pregnancies continue to have greater pelvic mobility even when not pregnant, suggesting that the effects of multiple pregnancies last even after childbirth. Additionally, the study noted that previous pain can make women more sensitive to pain later. This is important for late mothers because the combined effects of multiple pregnancies and increased pain sensitivity may make them more likely to experience pelvic girdle pain. These findings show that parity (number of previous pregnancies) plays an important role in the development of pelvic girdle pain, especially for late mothers.

One study [27] found that regular exercise during pregnancy had positive effects on late mothers (aged ≥35 years). Women in this group who exercised at least twice a week experienced reduced pelvic girdle pain and had lower gestational weight gain compared to those with less frequent physical activity. Furthermore, regular exercise was associated with a reduced risk of macrosomia (having a large baby), with fewer women in the exercise group experiencing gestational weight gain of 16 kg or more. These findings suggest that exercise in late pregnancy may help alleviate common pregnancy-related complications and promote better maternal health, supporting the importance of staying active for late mothers [27].

2.Psychological factors associated with low back pain in late mothers

A study [25] reported that older women (≥35 years) experienced fewer pregnancy-related symptoms compared to younger women (<25 years) and were also less likely to develop depression. This suggests that older mothers may be less psychologically affected by common pregnancy-related issues such as backache or nausea. However, while depression rates were lower, older mothers often reported stress related to other physical symptoms, including hemorrhoids and varicose veins, which are more common in older women during pregnancy. These physical discomforts may contribute to heightened emotional distress, highlighting that while older mothers may not suffer from clinical depression as often as their younger counterparts, they are still vulnerable to emotional challenges arising from physical discomfort [24].

Additionally, a study [27] found that older mothers, particularly those with increased parity, are at a higher risk for developing pelvic girdle pain (PGP), which could significantly affect their psychological well-being. Chronic pain, such as PGP, is often associated with increased emotional distress, including feelings of frustration, anxiety, and stress, which could negatively impact a woman’s overall mental health. Even though older mothers may experience fewer psychological complaints overall, the persistent physical discomforts associated with pregnancy-related pain are likely to influence their emotional state and coping mechanisms [26].

Furthermore, a study [27] emphasized the psychological benefits of regular exercise during pregnancy. Older mothers who engaged in physical activity were found to experience less pelvic girdle pain and lower gestational weight gain, which, in turn, reduced psychological distress. Exercise is known to promote the release of endorphins, which enhance mood and reduce anxiety, making it a valuable coping strategy for emotional well-being. These findings indicate that maintaining a healthy level of physical activity may help mitigate both the physical and psychological challenges that late mothers face during pregnancy [27].

3.Social factors influencing low back pain in late mothers

Social factors play a significant role in shaping how older mothers experience and manage low back pain during pregnancy. Access to healthcare, social support networks, and occupational responsibilities all contribute to their ability to cope with pain effectively.

Older mothers tend to use antenatal healthcare services differently than younger mothers. A study [24] found that late mothers had fewer antenatal visits and overnight hospital stays, which may contribute to delayed pain management for pregnancy-related discomforts like low back pain. This could be due to their confidence in managing symptoms independently, work commitments, or prior pregnancy experiences. However, a study [15] emphasized that timely screening and early intervention are crucial in preventing the progression of low back pain, especially for primiparous mothers. Insufficient engagement with healthcare professionals may lead to suboptimal guidance on pain management strategies, affecting older mothers’ overall well-being.

Social support, particularly from partners, family members, and healthcare providers, significantly influences how older mothers cope with low back pain. A study [24] highlighted that partner involvement and emotional support can reduce stress, which in turn can alleviate pain perception. Women with strong family support tend to receive physical assistance (e.g., help with daily tasks) and emotional reassurance, leading to better pain tolerance. Conversely, mothers with limited social support may experience higher stress levels, which can exacerbate physical discomfort.

Healthcare providers play a crucial role in offering practical guidance for pain management. A study [27] found that women who engaged in regular exercise during pregnancy reported lower levels of pregnancy-related pain, including pelvic girdle pain. However, older mothers may be less likely to seek professional guidance or participate in prenatal exercise programs due to time constraints or lack of awareness. Encouraging healthcare engagement and tailored exercise programs could significantly improve their pain outcomes.

Occupational factors also contribute to pregnancy-related back pain. A study [15] found that prolonged standing, heavy lifting, and sedentary work are key risk factors for low back pain in pregnancy. Many older mothers, particularly those with established careers, may find it difficult to modify their workload or take extended leave, increasing their physical strain. In addition, another study [26] emphasized that parity influences pain levels. Multiparous older women may experience intensified pelvic girdle pain, which can interfere with their ability to perform daily activities. This highlights the importance of workplace adjustments, ergonomic support, and structured maternity leave policies to accommodate the physical demands of pregnancy.

Similarly, a study [25] found that women who engaged in high-impact sports or regular physical activity before pregnancy had lower risks of pelvic girdle pain. This suggests that older mothers with a history of an active lifestyle may have better musculoskeletal health and experience less severe pregnancy-related pain. Encouraging safe physical activity adaptations during pregnancy could help mitigate low back pain symptoms.

## 5. Discussion

This systematic review highlights the complexity of low back pain (LBP) in older mothers and underscores the necessity of a biopsychosocial (BPS) approach for effective management. The findings align with the literature, emphasizing the interaction between biological, psychological, and social factors in contributing to and sustaining LBP in this population [5,16]. In the included studies, the state of well-being was primarily assessed using subjective measures such as self-reported pain intensity (e.g., Visual Analogue Scale), physical function questionnaires, and quality of life indicators related to physical discomfort.

### 5.1. Biological Factors

The biological dimension of LBP in older mothers is multifaceted, often exacerbated by pregnancy-related musculoskeletal changes and age-related degenerative processes. Late pregnancy has been recognized as a risk factor for pregnancy-related LBP due to increased spinal loading and hormonal changes affecting ligament laxity [15]. Additionally, degenerative changes in the facet joints, a known contributor to chronic LBP, tend to manifest more prominently in older individuals, further predisposing older mothers to pain [6]. These findings support prior research emphasizing the significant role of age-related spinal degeneration in chronic pain development [5]. Given these biological risk factors, non-pharmacological interventions such as physiotherapy (including manual therapy, e.g., spinal mobilization), tailored exercise regimens (e.g., pelvic stabilization exercises), aquatic therapy (e.g., water-based strengthening or stretching exercises), and massage and rehabilitation programs were highlighted across reviewed studies as crucial treatment modalities [2].

### 5.2. Psychological Factors

Psychological distress, including anxiety, depression, and catastrophizing, has been shown to exacerbate pain perception and disability among individuals with LBP [16]. Older mothers often experience heightened stress due to concerns about their maternal role, career demands, and societal stigma [11]. However, a study [24] found that despite experiencing physical discomfort, older mothers were significantly less likely than younger mothers to report symptoms such as frequent vomiting, back pain, and depression. Older women may have greater emotional regulation and a higher pain threshold due to previous life experiences, making them less likely to perceive or report common pregnancy-related discomforts as distressing. Additionally, they may be more accepting of pregnancy-related changes, viewing them as a natural and expected part of the maternal journey [23].

The psychological burden of LBP is further compounded by sleep disturbances, which can both stem from and exacerbate chronic pain, creating a reciprocal cycle that undermines physical recovery [2]. While sleep disturbances were not uniformly assessed across the included studies, related outcomes such as rest quality and daily functioning were indirectly addressed. Moreover, the tendency of patients to equate imaging findings with pain severity can lead to heightened fear-avoidance behaviors, ultimately prolonging disability [6]. These findings reinforce the necessity of cognitive-behavioral interventions to address maladaptive pain beliefs and improve pain coping strategies in older mothers, complemented by physiotherapy interventions such as tailored exercise programs, manual therapy, aquatic therapy (e.g., aquagym), and massage.

### 5.3. Social Factors

Social determinants of health, including societal perceptions of late motherhood, work–life balance challenges, and access to healthcare, significantly influence LBP outcomes. Older mothers may face stigma and judgment regarding their parental capabilities and the perceived risks associated with late pregnancy, contributing to psychological distress and pain persistence [10]. Additionally, the increasing trend of delayed motherhood, particularly among women with higher education and middle-class backgrounds, suggests that occupational and financial factors may also play a role in shaping their pain experiences [6]. Workplace accommodations and social support networks are, therefore, crucial in mitigating the burden of LBP in this population.

### 5.4. Clinical Implications

Given the multifaceted nature of LBP in older mothers, clinical management should incorporate multidisciplinary BPS interventions that address biological, psychological, and social contributors. Physiotherapy and exercise therapy should be complemented by psychological interventions, including cognitive-behavioral therapy, to improve pain coping mechanisms [2,19]. Moreover, targeted public health initiatives are needed to address social stigma and ensure equitable access to healthcare resources for older mothers.

Expanding on interdisciplinary contributions, physiotherapists, midwives, family physicians, and psychologists each bring essential perspectives to biopsychosocial care. For example, evidence suggests that BPS interventions led by trained physiotherapists in primary care improve disability and pain outcomes, especially when psychosocial elements like unhelpful thoughts and coping styles are prioritized and providers receive proper training and supervision (van Erp et al., 2018) [19]. Similarly, midwives may encounter ambivalence and trust issues when addressing pregnancy-related pelvic pain, highlighting the importance of empathetic communication and belief in patient narratives (Mogren et al., 2010) [28]. Psychologists working in perinatal health also play a vital role in mitigating healthcare disparities, advocating for patients, and addressing cultural beliefs that shape health behavior and pain expression (Stratmann, 2023) [29].

Although these studies are not specific to older mothers, they reflect effective interdisciplinary BPS strategies that are transferable and should be adapted to support this unique population. Additionally, cultural contexts, such as religious beliefs or socio-economic conditions, may influence how older mothers perceive and manage LBP and should be considered when designing holistic interventions (Rodrigues-de-Souza et al., 2016) [30].

It is also essential to distinguish between findings and null results in pain studies. For instance, psychological resilience, despite experiencing physical pain, has been observed in some studies [25]. This suggests that psychological factors might buffer against the impact of pain, which is significant for older mothers who might be particularly resilient due to their life experiences.

### 5.5. Strengths and Limitations

This review demonstrates a strong emphasis on the biopsychosocial factors influencing low back pain in older mothers, a population that so far has received limited research attention. A systematic methodology was employed, synthesizing findings from both cross-sectional and prospective cohort studies to ensure a comprehensive analysis. Rigorous inclusion and exclusion criteria were applied, particularly in selecting studies that explicitly focused on older mothers, thereby enhancing the relevance of the findings. Additionally, the review highlights the importance of understanding pain experiences through a biopsychosocial lens, contributing to a more holistic perspective on low back pain management.

While this systematic review provides valuable insights into the biopsychosocial dimensions of LBP in older mothers, certain limitations should be acknowledged. The review’s exclusion of studies published between January 2010 and May 2024 and those not in English may have restricted the scope of findings. Moreover, the reliance on observational studies restricts the ability to establish causal relationships between biopsychosocial factors and pain outcomes. This introduces selection bias, as observational studies may not fully capture the experiences of all older mothers with LBP, especially those who do not seek care due to socio-economic barriers, stigma, or other factors. This bias could limit the generalizability of the findings.

Furthermore, the absence of longitudinal data or intervention trials is a significant limitation. Longitudinal studies could provide insights into how biopsychosocial factors influence the progression of LBP over time, while intervention trials would help identify effective strategies for managing LBP in this population. The lack of such studies underscores the need for further research to establish causality and assess the long-term impact of biopsychosocial interventions.

In addition, heterogeneity in study designs, sample characteristics, and outcome measures may limit the comparability of findings. Despite using validated tools, the use of different assessment instruments across studies limits the ability to synthesize findings consistently. Nonetheless, these limitations underscore the need for future research to prioritize longitudinal studies and culturally sensitive approaches to deepen the understanding of biopsychosocial influences on low back pain in this population.

## 6. Conclusions

This systematic review synthesized evidence on the biopsychosocial (BPS) determinants of low back LBP in older mothers, addressing a critical research gap in maternal health. The findings demonstrate that while older mothers may report fewer overt psychological symptoms such as depression compared to younger women, the cumulative effects of age-related physical discomfort, such as pelvic girdle pain, hemorrhoids, and varicose veins, can still contribute to considerable psychological distress. Social and lifestyle factors, including limited healthcare utilization, lack of social support, and occupational demands, further influence the pain experience and coping strategies within this population.

These insights underscore the importance of early, multidisciplinary interventions that integrate physical therapies, psychosocial support, and workplace accommodations. Tailored exercise programs and empathetic counseling approaches may help alleviate the physical and emotional burdens associated with late motherhood.

Given the current reliance on observational studies, there is a pressing need for longitudinal research and clinical trials evaluating the effectiveness of BPS-based interventions in this demographic. Future efforts should also focus on developing screening tools and preventive strategies tailored to older pregnant individuals, taking into account cultural and contextual influences on health behavior and pain expression.

By aligning with the study’s original objective to explore how late motherhood is associated with LBP from a biopsychosocial perspective, this review contributes a nuanced understanding of the interconnected factors affecting older mothers’ well-being and calls for more inclusive, evidence-based maternal care strategies.

## Figures and Tables

**Figure 1 healthcare-13-01237-f001:**
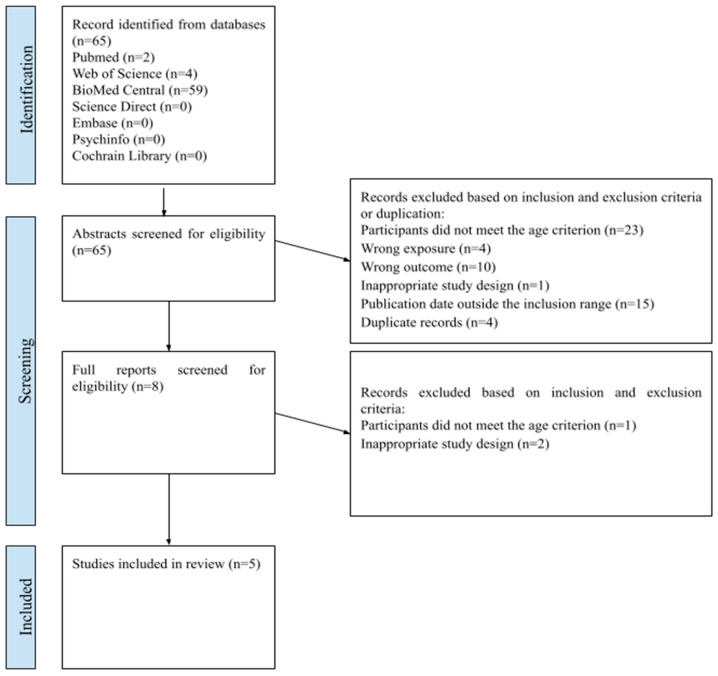
PRISMA flow diagram of the literature screening process.

**Table 1 healthcare-13-01237-t001:** Eligibility criteria (PICOS model).

Criterion	Description
Participants (types of participants)	Women aged 35 years or older who have experienced late motherhood (e.g., delayed childbirth, geriatric pregnancy, or motherhood after 35 years) and report low back pain during pregnancy (chronic low back pain persisting for more than 12 weeks, lumbar pain, lumbar spine pain, or nonspecific low back pain, defined as pregnancy-related lumbar discomfort not linked to a clearly identifiable medical condition).
Exposure of interest (independent variable)	Experiences related to late motherhood and their association with low back pain.
Comparator/Control	Studies may compare different groups within observational settings, such as older mothers vs. younger mothers, varying severity of low back pain, or different lifestyle factors.
Outcomes (dependent variable)	Biopsychosocial aspects related to low back pain, including physical (pain severity, disability), psychological (emotional distress, depression, anxiety), and social factors (work impact, social support).
Study type	Inclusion of observational studies, comprising cohort, cross-sectional, and case-control designs. Studies published in English between 2010 and May 2024.

**Table 2 healthcare-13-01237-t002:** Assessment of risk of bias in studies.

Study	Sample Size (N)	Design	Sampling Method	Risk of Bias (JBI)
Klemetti et al. [24]	2825	Cross-sectional study	Stratified random sampling via national birth registry	Moderate—sampling was well described, but the validity of the outcome measure was unclear
Owe et al. [25]	39,184	Cohort study	Voluntary population-based cohort recruitment	Low—large, population-based cohort with valid and reliable measurement of exposure and outcome; confounders identified and adjusted for; high follow-up rate; and appropriate statistical analysis
Bjelland et al. [26]	75,939	Cohort study	Consecutive sampling during routine prenatal care within a population-based cohort	Low—large, well-described cohort with validated recruitment methods; high response rate; pre-defined outcome measure; and appropriate adjustment for confounders in analysis
Haakstad et al. [27]	466	Cohort study	Convenience sampling with eligibility criteria	Moderate—prospective cohort with validated measures; confounders adjusted, but some attrition and retrospective exposure data introduce bias
Nazari et al. [15]	550	Cross-sectional study	Stratified random sampling with eligibility criteria	Moderate—clear objectives and methods; large sample, but causality cannot be inferred due to cross-sectional design

**Table 3 healthcare-13-01237-t003:** Study characteristics and key findings.

Author (Year)	Country	Study Design	Sample Size (N)	Population Characteristics	Intervention/ Exposure	Outcomes Measured	Key Findings	Risk of Bias
Klemetti et al. [24]	United Kingdom	Cross-sectional study	2825	Women at three months postpartum, categorized by age groups (<25, 25–34, and 35+)	No intervention (observational study)	Maternal health, socioeconomic status, and healthcare service use	Older women overall were significantly less likely to report repeated vomiting, backache, and experiencing depression com-pared with the youngest women. Proportion of women reporting health problems during pregnancy—backache: 65.3%—<25 49.5%—25–34 43.6%—35+.	Moderate
Owe et al. [25]	Norway	Prospective population-based cohort study	39,184	Women divided into age groups (<25, 25–29, 30–34, and ≥35 years)	No intervention (observational study)	Pelvic girdle pain, BMI, education, smoking, marital status, health history	Younger age, higher BMI, lower education, smoking, and a history of low back pain or depression were associated with a higher risk of pelvic girdle pain during pregnancy.	Low
Bjelland et al. [26]	Norway	Prospective cohort study	75,939	Women with different age groups, including younger (<25), middle age (25–34), and older (≥35)	No intervention (observational study)	Maternal age, BMI, parity, education, history of low back pain, emotional distress, physically demanding work, smoking in pregnancy, prepregnancy physical activity weekly	In the study, pelvic girdle syndrome prevalence was linked to maternal age. Women aged <25 had a 16.1% prevalence of pelvic girdle pain, while 2.8% had severe pain. In women aged 25–34, the prevalence was 15% for pelvic girdle pain and 2.5% for severe pain. For women aged ≥35, the prevalence was 14.4% and 2.3% for severe pain. Younger women had a slightly higher risk of pelvic pain.	Low
Haakstad et al. [27]	Norway	Prospective cohort study	466	Women divided into two groups: <35 years and ≥35 years	Regular exercise (≥2 times/week)	Prepregnancy BMI, lifestyle factors, gestational weight gain, pregnancy complaints, pelvic girdle pain, urinary incontinence, mode of delivery, section, instrumental delivery, vaginal birth, and newborn outcomes: birth weight, low birth weight, macrosomia (>4000 g), preterm birth, post-term birth	In women with advanced maternal age, exercising ≥2 times weekly was associated with less pelvic girdle pain (40.0% vs. 61.1%). 58.8% of women <35 years and 51.0% of women ≥35 years reported experiencing pelvic girdle pain during pregnancy. The difference in prevalence between age groups was not statistically significant (*p* = 0.15).	Moderate
Nazari et al. [15]	Iran	Cross-sectional study	550	Women divided into four groups by age: <20, 20–25, 26–30, and >30	No intervention (observational study)	Prevalence of low back pain (LBP), risk factors: maternal age, BMI, gestational age, duration of sitting and standing	The results showed a pregnancy-related LBP prevalence of 67.27%. In addition, the study of the factors influencing pregnancy-related LBP revealed that maternal age, gestational age, high BMI, and the ability to stand and sit only for less than 3 h were the most important risk factors. In the age groups, women aged 26–30 and over 30 had a higher percentage of low back pain (LBP) compared to younger women (<20 years and 20–25 years).	Moderate

## Data Availability

All relevant data are contained within the manuscript and its Appendix A.

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
