# Peer review of "Biopsychosocial Strategies for Alleviating Low Back Pain in Late Mothers: A Systematic Review"

_healthcare, 2025, doi:10.3390/healthcare13111237_

Round 1

Reviewer 1 Report

Comments and Suggestions for Authors

Reviewer Comments:

The topic is interesting and can bring benefits to the scientific community.

  1. Abstract
  • clarify and indicate the research problem in order to be able to address the study objective. The research question is not explicit
  • No relevant numerical data are included.
  • It does not directly link the findings to the research question.
  1. Introduction
  • The authors refer to low back pain, but knowing that there are differences between acute and chronic pain, it should be better contextualized when referring to the biopsychosocial model. My question is, would this model apply equally to acute and chronic pain?
  • It is not emphasized how this study fills a gap in the literature.
  1. Methods
  • The inclusion criteria regarding the type of low back pain should be specified. What time was considered in terms of chronic low back pain? what was considered non-specific? also neuropathic pain? Irradiated?
  • Clearly explain the sample selection criteria.
  1. Results
  • The flow chart is very simple. It should be developed making clear the number of articles found in the databases, explaining the reasons for inclusion and exclusion at each stage...
  • I suggest to improve the tables consistently for better understanding. Also the formatting
  1. Dicussion and Conclusion
  • Further develop the practical implications
  • Rephrase the conclusion to better connect with the objectives of the study. There is information that can be in the discussion section.

Author Response

Thank you for your valuable comments and suggestions to improve our manuscript. We have carefully addressed all your comments using the review function in the document. The revised manuscript includes all changes, which are highlighted and linked to the respective comments. In addition, we have provided detailed responses in the comment boxes and attached them as a separate Word file for your reference.

Answers for Reviewer 1

Comment#1

Abstract

clarify and indicate the research problem in order to be able to address the study objective. The research question is not explicit

No relevant numerical data are included.

It does not directly link the findings to the research question.

Answer: Thank you for your valuable feedback. In response to your comment regarding the clarity of the research problem and the absence of an explicit research question, we have revised the abstract to better reflect the study’s objective. Specifically, we have now included a clear statement that outlines the aim of the review-“to examine the biopsychosocial determinants of LBP in older mothers”-and elaborated on the focus of the analysis by specifying the biological, psychological, and social factors explored. While research questions are typically not stated in question form in structured abstracts, we have ensured that the scope and intent of the review are now clearly articulated. We have also included relevant numerical data (sample size: 118,964 participants) and strengthened the linkage between the findings and the study objective to address your concerns.

Comment#2

Introduction

The authors refer to low back pain, but knowing that there are differences between acute and chronic pain, it should be better contextualized when referring to the biopsychosocial model. My question is, would this model apply equally to acute and chronic pain?

Answer: Regarding the comment, we provided the information about chronic and acute pain in BPS model and its relevance for both types of pain. The recent inclusion of the study by Willson and colleagues (2023) highlights the established influence of biopsychosocial factors on various forms of chronic pain. However, their specific contribution to pain experienced during late pregnancy remains relatively underexplored, indicating a critical gap in the existing literature

It is not emphasized how this study fills a gap in the literature

Answer: We have added information about the importance of the study, stating that pain among older mothers is an unique experience that may have been inadequately addressed in literature.

Comment #3

Methods

The inclusion criteria regarding the type of low back pain should be specified. What time was considered in terms of chronic low back pain? What was considered non-specific? Also neuropathic pain? Irradiated?

Clearly explain the sample selection criteria.

Answer: We appreciate your observation. We have revised the methods section to include precise definitions of the types of low back pain considered in this review. Chronic low back pain was defined as pain persisting for more than 12 weeks. We included studies referring to nonspecific low back pain, defined as pregnancy-related lumbar discomfort not attributable to a clear medical diagnosis.

Comment #4

Results

The flowchart is very simple. It should be developed, making clear the number of articles found in the databases, explaining the reasons for inclusion and exclusion at each stage…

Answer: Thank you for your comment and the flow chart has bben developed based on your comments.

I suggest to improve the tables consistently for better understanding. Also the formatting.

Answer: Thank you for these constructive suggestions. In response, we have expanded the PRISMA flow diagram to present a detailed overview of the article screening process, including the number of records identified, screened, excluded (with reasons), and finally included in the analysis.

Comment #5

Discussion and Conclusion

Further develop the practical implications

Answer: Thank you for highlighting the need to further develop the practical implications of our findings. We have revised the Clinical Implications section in the Discussion to more clearly articulate how physiotherapists, midwives, family physicians, and psychologists can apply a biopsychosocial approach when supporting older mothers with low back pain. These revisions include specific examples from the literature and emphasize the value of interdisciplinary collaboration. We trust that this addresses your suggestion and strengthens the practical relevance of our review.

Rephrase the conclusion to better connect with the objectives of the study. There is information that can be in the discussion section.

Comment #6

Rephrase the conclusion to better connect with the objectives of the study. There is information that can be in the discussion section.

Answer: We thank Reviewer 1 for the insightful suggestion to rephrase the conclusion to better align with the study's objectives and to relocate content more appropriate for the discussion section. In response, we revised the conclusion to directly reflect the primary aim of the review, examining the biopsychosocial determinants of LBP in older mothers, and to succinctly summarize the key findings relevant to our research questions.

Reviewer 2 Report

Comments and Suggestions for Authors

It is an interesting topic.

Lines 35-36: ,,.. state that LBP is the primary cause of disability globally, with one in six Australians reporting back issues in 2017–2018….”

You mentioned disability related to low back pain in Australia in 2017-2018. Do you have data from other continents for the same period? Is it male or female? However, your research refers to female individuals and it would have been interesting if it was women (1 in 56) or just men.

Lines 40-43: ,,Non-pharmacological care is regarded as a primary treatment option, leading individuals with LBP to frequently seek assistance from physiotherapists and other rehabilitation health professionals for managing both short-term and long-term LBP [2].

In the Introduction chapter, do you refer in general to low back pain, namely to the possibility that people with low back pain can go directly to physiotherapists? Will they make the diagnosis and apply the treatment? Or should the rehabilitation doctor or a multidisciplinary team make the diagnosis and indicate the therapeutic conduct? “

Line 107: ,,..How does LBP impact the daily lives and well-being of older mothers? “

Please comment on what methods of assessing the state of well-being you have applied.

Lines 351-353:,,Given these biological risk 351 factors, non-pharmacological interventions such as physiotherapy and rehabilitation pro-352 grams remain crucial treatment modalities [2].”

What exactly are you referring to when you say physiotherapy for pregnant women? Massage? Physiotherapy? Aquagym?

In lines 385-386 you only refer to physiotherapy.

Lines 366-367: ,,The psychological burden of LBP is further compounded by sleep disturbances which are both a consequence of and a contributing factor to chronic pain [2]."

Do all 5 studies address the issue of sleep?

Lines 385-387: ,, Physiotherapy and exercise therapy should be complemented by phychological interventions, including cognitive-behavioral therapy, to improve pain coping mechanisms [2,16]".

 Are you reffering to only  kinetotherapy? When you say physiotherapy what do you mean? You mentioned the limitations of the study.

It seems like a well-organized paper.

I appreciate the work done for this study.

My comments are only intended to make the paper better. Good luck!

Author Response

Thank you for your valuable comments and suggestions to improve our manuscript. We have carefully addressed all your comments using the "Review function" in the document. The revised manuscript includes all changes, which are highlighted and linked to the respective comments. In addition, we have provided detailed responses in the comment boxes and attached them as a separate Word file for your reference.

Answers for Reviewer 2

Comment #1

Lines 35-36: ,,.. state that LBP is the primary cause of disability globally, with one in six Australians reporting back issues in 2017–2018….”

Answer: We appreciate your valuable remarks. Wu and colleagues study (2020) indications have been extensively elaborated in order to showcase the global situation of this phenomenon.

Comment #2

You mentioned disability related to low back pain in Australia in 2017-2018. Do you have data from other continents for the same period? Is it male or female? However, your research refers to female individuals and it would have been interesting if it was women (1 in 56) or just men.

Answer: Thank you for your comment. Beyond Australian data, Wu et al. (2020) found significant global variation in the prevalence of low back pain (LBP). In 2017, Southern Latin America had the highest prevalence at 13.47%, followed by high-income Asia Pacific at 13.16%. East Asia reported the lowest prevalence at 3.92%, with Central Latin America slightly higher at 5.62%. In terms of the number of affected individuals, South Asia bore the highest burden with 96.3 million cases, while Oceania and the Caribbean reported the lowest at 0.7 million and 2.7 million respectively. According to the World Health Organization (2023), LBP is defined as pain between the lower ribs and buttocks and can be acute, sub-acute, or chronic in duration.

Comment #3

Lines 40-43: ,,Non-pharmacological care is regarded as a primary treatment option, leading individuals with LBP to frequently seek assistance from physiotherapists and other rehabilitation health professionals for managing both short-term and long-term LBP [2].

Kindly see the comment below addressing the problem.

Comment #4

In the Introduction chapter, do you refer in general to low back pain, namely to the possibility that people with low back pain can go directly to physiotherapists? Will they make the diagnosis and apply the treatment? Or should the rehabilitation doctor or a multidisciplinary team make the diagnosis and indicate the therapeutic conduct? “

Answer: Thank you for your observation. In the revised version, we clarify that while individuals with low back pain (LBP) often seek non-pharmacological care, including physiotherapy, the diagnostic and therapeutic process should ideally involve a coordinated approach. The role of the rehabilitation physician or a multidisciplinary team remains central in making the initial diagnosis and determining the overall treatment strategy. Physiotherapists and other health professionals then implement the care plan within this framework. This clarification ensures alignment with best practices in clinical pathways for LBP management and acknowledges the importance of interdisciplinary collaboration in both diagnosis and treatment.

Comment #5

Line 107: ,,..How does LBP impact the daily lives and well-being of older mothers? “

Answer:Thank you for your comments. A systematic review was incorporated to highlight how the prevalence of low back pain varies across different age groups. Dionne (2006) conducted a comprehensive review which revealed that the occurrence of severe back pain tends to increase with advancing age. This suggests that aging is a significant factor contributing to the burden of low back pain. Thus, this systematic review provides valuable insight into the age-related patterns of LBP, emphasizing the need for age-specific prevention and management strategies.

Moreover, a newly integrated study by Willson and colleagues(colleagues (2023) revealed that Pain is experienced differently among individuals during pregnancy, and women scheduled for cesarean delivery-an increasingly common procedure-represent a relatively understudied group who may be at increased risk of pain.

Comment#6

Please comment on what methods of assessing the state of well-being you have applied.

Answer: Thank you for pointing this out. To evaluate these non-pharmacological interventions, the well-being in the included studies was primarily assessed using subjective measures. These included self-reported pain intensity, often measured by tools like the Visual Analogue Scale (VAS), which allows participants to indicate their pain levels on a scale. In addition, physical function questionnaires were used to evaluate the degree of disability or functional limitations caused by LBP, and quality-of-life indicators related to physical discomfort and emotional well-being were assessed. These measures are essential in understanding how interventions impact the daily activities and overall health of late mothers experiencing LBP.

Comment#7

Lines 351-353:,,Given these biological risk 351 factors, non-pharmacological interventions such as physiotherapy and rehabilitation pro-352 grams remain crucial treatment modalities [2].”

Answer: Thank you for your insightful comment. In the context of alleviating LBP in late mothers, "physiotherapy" encompasses a range of non-pharmacological interventions, specifically designed to address pregnancy-related musculoskeletal changes and age-related degenerative processes. These interventions may include manual therapy, tailored exercise regimens (such as pelvic stabilization exercises), aquatic therapy (Aquagym), and massage therapy. These approaches have been shown to provide relief for LBP by improving spinal mobility, reducing pain, and enhancing overall physical well-being during late pregnancy, without relying on pharmacological treatments that may present risks to the mother or fetus.

Comment#8

What exactly are you referring to when you say physiotherapy for pregnant women? Massage? Physiotherapy? Aquagym?

Answer: Thank you for your continued thoughtful feedback. By “physiotherapy,” we refer to a range of evidence-based, non-pharmacological interventions tailored to the musculoskeletal needs of pregnant women. These include manual therapy (e.g., spinal mobilization), tailored exercise regimens (e.g., pelvic stabilization), aquatic therapy (aquagym), and massage therapy. These approaches aim to alleviate pain, enhance mobility, and improve overall physical function. This clarification has been added to lines 389–392.

Comment#9

In lines 385-386 you only refer to physiotherapy.

Answer: Thank you for your valuable feedback. In response to your query, we have clarified the term physiotherapy to include evidence-based non-pharmacological interventions such as manual therapy, tailored exercise regimens (e.g., pelvic stabilization), aquatic therapy (aquagym), and massage therapy. These approaches are now specified in lines 389–392. (Please refer to those lines)

Comment #10

Lines 366-367: ,,The psychological burden of LBP is further compounded by sleep disturbances which are both a consequence of and a contributing factor to chronic pain [2]."

Answer: Answer to comment 12 is applicable

Comment #11

Do all 5 studies address the issue of sleep?

Answer: Answer to comment 12 is applicable

Comment #12

Lines 385-387: ,, Physiotherapy and exercise therapy should be complemented by phychological interventions, including cognitive-behavioral therapy, to improve pain coping mechanisms [2,16]".

Answer: Thank you for your insightful observations. In response, we have clarified our use of the term "physiotherapy" to reflect a comprehensive set of non-pharmacological physical treatments, including manual therapy, tailored exercises (e.g., pelvic stabilization), aquatic therapy (aquagym), and massage. We also elaborated on the psychological dimension of chronic LBP, highlighting how cognitive-behavioral therapy can support improved coping. Regarding sleep disturbances, we acknowledge that while not all included studies assessed sleep outcomes directly, some addressed related aspects such as rest quality or functioning. These clarifications have been incorporated into lines 419–429 of the revised manuscript.

Comment# 13

Are you reffering to only  kinetotherapy? When you say physiotherapy what do you mean? You mentioned the limitations of the study.

Answer: Thank you for your helpful comment. In our review, we use the term "physiotherapy" to refer broadly to non-pharmacological physical interventions, including but not limited to-kinetotherapy, manual therapy, tailored exercise programs, aquatic therapy, and massage. We have clarified this in the relevant sections of the manuscript. We also acknowledge in the limitations that while these modalities are commonly reported, variations in terminology and intervention protocols across studies may limit direct comparability.

Reviewer 3 Report

Comments and Suggestions for Authors

Dear authors,

We sincerely appreciate your commitment, effort, and time dedicated to advancing research in the study population you have examined, as well as your attempt to incorporate complementary methodologies. Below, I share my observations, which I hope will be helpful and provide constructive guidance.

The manuscript contributes to a relatively unexplored but highly relevant thematic intersection: low back pain (LBP) in the context of late motherhood. The use of the biopsychosocial (BPS) model to structure and interpret the findings is appropriate. However, the manuscript could benefit from greater methodological and interpretive depth, particularly in relation to bias analysis, quality assessment of the included studies, and the framing of the conclusions.

1. Title and Abstract
• The title accurately reflects the content.
• The abstract is informative, though it could be improved by clearly quantifying key findings and indicating how many studies addressed each component of the BPS model.

2. Introduction
• The introduction effectively contextualizes the significance of LBP in older mothers and the relevance of the BPS approach.
• It is recommended to reinforce the connection between the increase in maternal age and the rising epidemiological prevalence of low back pain in this population.

3. Methodology
There are several methodological aspects to consider:

• It would be advisable to specify the end date of the literature search.
• To ensure replicability, the full search strategy should be included (as supplementary material), including Boolean logic, MeSH terms, filters, and results per database.
• It is necessary to clarify whether a formal tool was used to assess risk of bias and to provide the corresponding results.
• Although a narrative synthesis is appropriate given the heterogeneity of the studies, it would be valuable to include a structured table showing methodological quality scores, following the aforementioned criteria.

4. Results
There are some areas I consider relevant to highlight:

• Discuss the impact of variability in sample sizes on the strength of the evidence.
• Add a table including methodological quality, risk of bias, sampling methods, and adjustment for confounding variables.
• It remains unclear which studies contribute evidence to each specific research question.

5. Discussion

Although the integration of findings across the three domains of the BPS model is well accomplished, and references are appropriately incorporated, I offer the following suggestions:

• The clinical implications should be expanded for physiotherapists, midwives, family physicians, and other members of the transdisciplinary team.
• More explicitly distinguish between findings and null results (e.g., psychological resilience despite physical pain).
• Could the authors consider how cultural factors may influence the experience and expression of low back pain?
• Regarding the limitations, while linguistic and temporal restrictions are acknowledged, the discussion of selection bias—stemming from the inclusion of observational studies only—could be expanded.
Additionally, the absence of longitudinal data or intervention trials deserves further reflection.

6. Conclusions

The conclusions are consistent with the findings and are cautiously formulated. I would suggest proposing specific avenues for future research (e.g., clinical trials with BPS-based interventions) and considering the development of screening tools or preventive strategies tailored to older pregnant individuals.

Author Response

Thank you for your valuable comments and suggestions to improve our manuscript. We have carefully addressed all your comments using the "Review" function in the document. The revised manuscript includes all changes, which are highlighted and linked to the respective comments. In addition, we have provided detailed responses in the comment boxes and attached them as a separate Word file for your reference.

Answers for Reviewer 3

Comments and Suggestions for Authors

Dear authors,

We sincerely appreciate your commitment, effort, and time dedicated to advancing research in the study population you have examined, as well as your attempt to incorporate complementary methodologies. Below, I share my observations, which I hope will be helpful and provide constructive guidance.

The manuscript contributes to a relatively unexplored but highly relevant thematic intersection: low back pain (LBP) in the context of late motherhood. The use of the biopsychosocial (BPS) model to structure and interpret the findings is appropriate. However, the manuscript could benefit from greater methodological and interpretive depth, particularly in relation to bias analysis, quality assessment of the included studies, and the framing of the conclusions.

Comment #1

  1. Title and Abstract
  • The title accurately reflects the content.
  • The abstract is informative, though it could be improved by clearly quantifying key findings and indicating how many studies addressed each component of the BPS model.

Answer: Thank you for your feedback. In response to your comment, we have revised the abstract to clearly quantify the key findings. Specifically, all five studies addressed biological factors (e.g., pelvic girdle pain, varicose veins), three studies focused on psychological factors (e.g., depression, stress), and two studies examined social factors (e.g., social support, healthcare access). This revision provides a clearer breakdown of how each component of the BPS model was addressed in the studies.

Comment #2

  1. Introduction
  • The introduction effectively contextualizes the significance of LBP in older mothers and the relevance of the BPS approach.
  • It is recommended to reinforce the connection between the increase in maternal age and the rising epidemiological prevalence of low back pain in this population.

Answer: In response, we have revised the Introduction to explicitly emphasize the link between the rising trend of delayed motherhood and the increasing global incidence of low back pain.

A systematic review was incorporated to highlight how the prevalence of low back pain varies across different age groups. Dionne (2006) conducted a comprehensive review which revealed that the occurrence of severe back pain tends to increase with advancing age. This suggests that aging is a significant factor contributing to the burden of low back pain. Thus, this systematic review provides valuable insight into the age-related patterns of LBP, emphasizing the need for age-specific prevention and management strategies.

Comment #3

  1. Methodology

There are several methodological aspects to consider:

  • It would be advisable to specify the end date of the literature search.
  • To ensure replicability, the full search strategy should be included (as supplementary material), including Boolean logic, MeSH terms, filters, and results per database.

Answer: In response to the reviewer’s suggestion, we have included the full search strategy as supplementary material. This includes details such as end dates of the literature search, Boolean operators, MeSH terms, applied filters, and the number of results retrieved from each database

  • It is necessary to clarify whether a formal tool was used to assess risk of bias and to provide the corresponding results.

Answer: Thank you for your valuable comment. In response, we have clarified that the JBI Critical Appraisal Checklists were used to assess the risk of bias in the included studies.  The results of the risk of bias assessments have been summarized and are now presented in Table 2.

Although a narrative synthesis is appropriate given the heterogeneity of the studies, it would be valuable to include a structured table showing methodological quality scores, following the aforementioned criteria.

Answer: Thank you for your valuable comment regarding the risk of bias assessment. In response, we have added a dedicated subsection titled "Risk of Bias Assessment" to clarify our approach. We appreciate your suggestion and have also included a structured table summarizing the methodological quality scores for each study, following the JBI Critical Appraisal Checklist. This table provides a clear overview of the risk of bias for each study, based on the suggested criteria, enhancing the transparency and robustness of our synthesis.

Comment #4

  1. Results

There are some areas I consider relevant to highlight:

  • Discuss the impact of variability in sample sizes on the strength of the evidence.
  • Add a table including methodological quality, risk of bias, sampling methods, and adjustment for confounding variables.

Answer: Thank you for your comment. We have clarified the procedure used to assess the risk of bias in the included studies. Specifically, we used the JBI Critical Appraisal Checklist, which provides a structured evaluation of the methodological rigor of each study. We considered various factors, including study design, sample selection, outcome measurement, statistical analysis, and potential sources of bias. Studies were categorized into low, moderate, or high risk based on the quality of their methodological features. We have created a table that includes sample size, study design, and sampling method. The sample size reflects the scale and power of each study, indicating how well the findings can be generalized to the target population. The sampling method provides insight into how participants were selected, influencing the representativeness and potential biases of the sample. A larger sample size generally increases the statistical power of a study, while the sampling method determines the level of randomness and control over potential confounding variables.

  • It remains unclear which studies contribute evidence to each specific research question.

Answer: Thank you for your insightful comment. This systematic review was guided by the biopsychosocial (BPS) framework to explore how late motherhood is associated with low back pain (LBP), with specific attention to biological, psychological, and social contributors, as well as the impact on daily life and well-being. We acknowledge that all five included studies (Owe et al., 2016; Nazari et al., 2020; Klemetti et al., 2011; Haakstad et al., 2019; Bjelland et al., 2010) primarily focused on biological factors. While none explicitly addressed psychological dimensions (e.g., emotional well-being or mental health), some studies indirectly noted the absence of these factors (e.g., Owe et al., 2016; Nazari et al., 2020), highlighting them as limitations. Social aspects, such as antenatal service use and age-related risks, were discussed in Klemetti et al. (2011), offering contextual insight into broader determinants of LBP. We have revised the manuscript to clearly attribute each study's contributions to the respective components of the BPS model and clarified these observations in the Results and Discussion sections.

Comment#5

  1. Discussion

Although the integration of findings across the three domains of the BPS model is well accomplished, and references are appropriately incorporated, I offer the following suggestions:

  • The clinical implications should be expanded for physiotherapists, midwives, family physicians, and other members of the transdisciplinary team.
  • More explicitly distinguish between findings and null results (e.g., psychological resilience despite physical pain).
  • Could the authors consider how cultural factors may influence the experience and expression of low back pain?

Answer: Thank you for your thoughtful and constructive comments. We have revised the clinical implications section based on your suggestions (Comment #5). Specifically:

  • We have expanded the clinical implications to include the roles of physiotherapists, midwives, family physicians, and other members of the transdisciplinary team, emphasizing their contributions to biopsychosocial (BPS) care for older mothers with low back pain (LBP).

  • We have incorporated a clearer distinction between findings and null results, particularly regarding psychological resilience despite physical pain, to ensure greater clarity in the interpretation of the results.

  • We have addressed the influence of cultural factors on the experience and expression of LBP, considering how religious beliefs, socio-economic conditions, and cultural contexts may shape the way older mothers perceive and manage their pain.

  • Regarding the limitations, we have included a more detailed discussion on the potential selection bias inherent in observational studies, as well as the absence of longitudinal data or intervention trials. These limitations are now clearly outlined in the limitations section to offer a more comprehensive understanding of the study's scope and areas needing further investigation.
  • Regarding the limitations, while linguistic and temporal restrictions are acknowledged, the discussion of selection bias—stemming from the inclusion of observational studies only—could be expanded.

Additionally, the absence of longitudinal data or intervention trials deserves further reflection.

Answer: Thank you for your thoughtful comment. In response, we have expanded the Strengths and Limitations section to address the potential selection bias associated with the inclusion of observational studies. We now note that such studies may not fully capture the experiences of older mothers who avoid seeking care due to stigma or socio-economic barriers. We have also elaborated on the absence of longitudinal data and intervention trials, emphasizing the need for future research that can evaluate the evolution of pain and psychosocial factors over time, and test the effectiveness of biopsychosocial interventions specifically tailored to this population.

Comment #6

  1. Conclusions

The conclusions are consistent with the findings and are cautiously formulated. I would suggest proposing specific avenues for future research (e.g., clinical trials with BPS-based interventions) and considering the development of screening tools or preventive strategies tailored to older pregnant individuals.

Answer: We appreciate your recommendation to propose specific avenues for future research. Accordingly, we have expanded the conclusion to emphasize the need for longitudinal studies, clinical trials evaluating BPS-based interventions, and the development of culturally sensitive screening tools and preventive strategies. These additions aim to strengthen the practical implications of our findings and guide future work in this field.

Round 2

Reviewer 1 Report

Comments and Suggestions for Authors

Most of the reviewer's contributions and suggestions have been addressed, although the format of the tables and figures for publication needs to be improved.

Reviewer 3 Report

Comments and Suggestions for Authors

Dear Authors,

I would like to begin this communication by thanking you for submitting the revised version of the manuscript entitled Biopsychosocial Strategies for Alleviating Low Back Pain in Late Mothers: A Systematic Review, as well as for the evident effort shown in incorporating the comments and suggestions arising from the peer review process.

I have carefully reviewed the current version of the article and can confirm that you have addressed the previously raised observations with rigor and depth. Particularly noteworthy are the following aspects: the expansion and updating of the introduction, the clarification of methodological procedures, and the acknowledgment of the study’s limitations.

These improvements have significantly enhanced the overall quality of the manuscript, both in terms of clarity of presentation and scientific consistency. The focus of the study is now more clearly articulated, and the contribution it makes to the disciplinary field is more evident and relevant.

In this regard, I would like to express my positive assessment of the submitted work and to congratulate you on the quality achieved in this final version.

I encourage you to continue your research endeavors and to keep contributing with studies that strengthen the development of scientific knowledge.

With kind regards,